# The Andalusian Registry of Donors for Biomedical Research: Five Years of History

**DOI:** 10.3390/biotech10010006

**Published:** 2021-03-12

**Authors:** Rocío Aguilar-Quesada, Inés Aroca-Siendones, Leticia de la Torre, Sonia Panadero-Fajardo, Juan David Rejón, Ana María Sánchez-López, Blanca Miranda

**Affiliations:** Andalusian Public Health System Biobank, Coordinating Node, 18016 Granada, Spain; arocasiendones@gmail.com (I.A.-S.); leticia.torre.ortega@juntadeandalucia.es (L.d.l.T.); sonia.panadero@juntadeandalucia.es (S.P.-F.); juand.rejon@juntadeandalucia.es (J.D.R.); anam.sanchez.exts@juntadeandalucia.es (A.M.S.-L.); bmiranda@friat.es (B.M.)

**Keywords:** biobanks sustainability, prospective collection, community engagement

## Abstract

The mission of the Andalusian Public Health System Biobank is to offer the best options for biological samples of human origin and associated clinical information, protecting the rights of citizens who donate their samples for research. Since the Andalusian Biobank provides high-quality biological samples of all types in a specified format, adapting the preanalytical phase according to the requirements of the research, prospective collection and distribution of samples are being prioritized in order to contribute to the sustainability of the Biobank. The Andalusian Registry of Donors for Biomedical Research is a tool for the recruitment of donors and the prospective collection of samples. Its operation is based on the informed consent of donors for their incorporation into the Registry and contact with possible donors under request from specific projects. An additional advantage of this unique initiative is to ensure that societal actors work together throughout the entire research process, establishing alliances with patient associations and groups to develop joint actions and promote biomedical research. Here, we describe the creation, ethical–legal aspects, management and results of the Andalusian Registry of Donors for Biomedical Research after five years of operation.

## 1. Introduction

The Andalusian Public Health System Biobank (SSPA Biobank) is an initiative of the Andalusian Regional Government of Health to promote biomedical research and protect the rights of citizens who donate their samples for research. It integrates any facility of the Andalusian Public Health System dedicated to the collection, processing, conservation and provision of human biological samples and clinical information for research, in addition to all blood and tissue centers for therapeutic purposes. The SSPA Biobank for research is organized as a network formed by multiple nodes in each Andalusian province working as a single decentralized structure, with a common ethical–legal and administrative framework and a comprehensive quality management system.

The SSPA Biobank handles all types of biological samples in the format requested from patients with different pathological conditions or from healthy donors, including derivatives related to liquid biopsies, tissues or living cell biobanking [1]. Precisely due to the multiple options of biospecimens to be analyzed corresponding to multiple diseases in the framework of research projects and their increase in complexity [2], it is unsustainable to have ready-to-use representative collections. On the other hand, the variability of biomarkers to be tested and the fact that the preanalytical phase must be carefully considered to ensure the integrity of these biomarkers [3] make fit-for-purpose procedures mostly necessary.

Alternatively, prospective collection is a model of providing biospecimens for research based on real-time requests, the requirements, advantages and disadvantages of which have been described [4]. The SSPA Biobank has incorporated this alternative model to the classic biobank operations to meet specific preanalytical requirements through innovative evidence-based standard operating procedures (SOPs), thus guaranteeing its mission of offering the researcher the best options for biological samples. Since prospective collection allows meeting the needs of the current research community, responding to research trends and also avoiding mistakes associated with the unrealistic utilization of biospecimens [5], it is being prioritized by the SSPA Biobank. The flexibility of this model makes it possible for the same donation to contribute to different research projects thanks to obtaining several derivatives, minimizing the sample volume and optimizing the use of resources as well as the trust of the donors in the management of the Biobank.

Within the framework of these objectives, and, at the same time, trying to involve societal actors in the research process, in 2015, the Andalusian Registry of Donors for Biomedical Research was created. It is an innovative and complementary tool for the recruitment of donors and the prospective collection of samples that respond to patients interested in donating samples for research. The purpose of this Registry of Donors is to have a database available that contains the population willing to donate samples and associated information for biomedical research. Furthermore, direct donor collaboration with the SSPA Biobank could have a positive impact in terms of usability on the sustainability of the samples.

Before its constitution and operation, ethical–legal bases were established through joint actions between the Andalusian Regional Ministry of Health and Ethics Committees and the SSPA Biobank. Given that the fundamental process of the Registry of Donors is the obtaining, recording and processing of the personal data of the participants and, in particular, sensitive information related to data concerning health, its creation was carried out in accordance with European and Spanish data protection regulations, specifically Regulation (EU) 2016/679 of the European Parliament and of the Council (General Data Protection Regulation), and Organic Law 3/2018 on the Protection of Personal Data and Guarantees of Digital Rights, respectively. In Andalusia, the Registry of Donors was legally established through regional regulations that identified the controller (Andalusian Regional Government of Health, responsible for the treatment) and the processor (SSPA Biobank, in charge of the treatment). Since a similar initiative was not previously known, the informed consent forms from the SSPA Biobank and from donor medical records such as the bone marrow registry were used as a reference for the elaboration of the specific informed consent form to incorporate into the Registry of Donors. This form explains the purpose of the Registry of Donors, how the data will be processed, the possibility of contact later to collect new data and invite participants to donate samples or data for a research project and also the option of revocation of the consent given. In relation to data processing, the donor’s signature on the informed consent form authorizes the SSPA Biobank to access and handle the personal data that are needed for the aims of the Registry of Donors (Table 1). Donors can exercise their right to access, correct, contest or remove their data, and any other rights recognized by current regulations, by contacting whoever is responsible for the treatment. Donors can raise any questions regarding the protection of their personal data with the SSPA Data Protection Officer.

## 2. Functioning, Database and Management System

The operation of the Andalusian Registry of Donors for Biomedical Research is based on the following processes shown in Figure 1: promotion; incorporation of participants into the Registry of Donors after the informed consent process; management of the donor database and access to medical records; contact with potential donors on request for a specific research project; and, finally, sample collection after a new and specific donor informed consent process.

### 2.1. Promotion of the Registry of Donors and Outreach Activities

Although the Andalusian Registry of Donors for Biomedical Research is aimed at the general population, the SSPA Biobank organizes different promotion and dissemination activities, adapting the approach and the language according to the audience: citizens, health workers and researchers. 

Given that citizenship is the fundamental basis of the Registry of Donors, numerous and inviting activities are organized annually to reach this audience and raise awareness about the importance of donating biological samples for biomedical research. Citizens receive information on the handling of biological samples for research and the operation of the SSPA Biobank, as well as the Registry of Donors. These activities include introductory talks online and on-site at the SSPA Biobank and educational facilities; scientific exhibitions and demonstrations at science fairs and educational centers; open-doors days and guided visits to the nodes of the SSPA Biobank for the educational community; exhibition stands in hospitals, science fairs or events and workshops organized with the participation of patient organizations or by different institutions related to health promotion such as sporting events. The SSPA Biobank also participates in both national and international scientific dissemination projects, such as the European Researchers’ Night, Science Week and Pint of Science (short talks organized in pubs), or commemorative days such as International Day of Women and Girls in Science or International Donor Day, among others. 

Ad hoc campaigns are also organized according to specific research priorities and target specific population groups. This was the case with the COVID-19 campaign for citizens who had overcome the disease. Other campaigns aimed at patient associations are permanent and consist mainly of workshops and information stands at the associations’ facilities and events organized by them on specific pathologies. In addition, it should be noted that patient associations are key partners to promote and to disseminate the Registry of Donors to the population, due to their deep awareness of the importance of collaboration with research projects and their role as active members of research [6,7]. Thus, periodically, collaborations are established between the SSPA Biobank and patient associations interested in this task.

Health workers are also key players in the Registry of Donors, as they are a reference for patients and are part of the sample collection process. For this reason, introductory talks and educational workshops are planned for them and for students of medical and nursing schools, taking advantage of events related to health. 

Finally, outreach activities of the Registry of Donors are prepared for researchers, as possible users of the biological samples donated by citizens previously registered in the Registry. To inform researchers, talks or stands are prepared to be presented at research institutions and biotechnology companies, networking events related to health and national and international conferences. 

In relation to the corporate image of the Andalusian Registry of Donors for Biomedical Research and as a support during the multiple activities detailed above, materials such as posters, flyers, brochures, roll-ups or comics have been designed. These materials are also useful tools at other times such as the informed consent process. A website and social media are also available. Additionally, press and television news have been launched with the main achievements and organized activities, advertising banners have been included on related websites with information from the Registry of Donors, short videos have been recorded and participation in radio and television broadcasts of regional channels has been promoted.

### 2.2. Registration of Donors

Incorporation into the Registry of Donors can be carried out through its website by authenticating the participants, through patient or citizen organizations that have established a prior agreement with the SSPA Biobank or directly through the SSPA Biobank staff.

Interested citizens must complete an application form that includes personal data and health-related information, as well as the specific informed consent form for the Registry of Donors. The information included in both forms is shown in Table 1. Once both forms are signed, the new potential donors are included in the Registry of Donors database, and a Welcome Letter signed by the Scientific Director of the SSPA Biobank, as well as a Card of Donor with identification and the donor number, is sent to the new participant.

### 2.3. Searching and Contact with Donors

When investigators request biological samples from the SSPA Biobank that are not available as part of the retrospective collections, possible donors are searched by the SSPA Biobank staff in the Register of Donors database using their available information. The selection is carried out using the requirements of the research project, such as the number of cases and the inclusion–exclusion criteria. Qualified personnel then access their medical records to verify that they meet the requirements. Validated donors are contacted by email or phone to give them information about the project objectives, type of samples and data that will be associated with donating, and to ask for their participation. In case they are willing to donate, an appointment will be provided to them for sample collection after the corresponding informed consent process. After donation, they are listed as unavailable for a 3-month downtime.

On the contrary, if the selected donors are not willing to participate in the research project, it is recorded in the database along with the specific reasons, and they remain available for future consultation.

When the number of validated donors does not reach the necessary number, the patient associations related to the scope of the project are contacted to request their collaboration with the SSPA Biobank for the promotion of the Register of Donors and donation of samples among their members. Alternatively, specific campaigns are organized as detailed above.

### 2.4. Sample Collection and Distribution

The collection site for obtaining the samples will depend on the type of sample and pathology, as well as the proximity and the conditions of mobility of the donors, and the availability of health and blood centers of the Andalusian Public Health System. Invasive samples are obtained by healthcare workers following standard clinical procedures. For non-invasive samples (nails, stool, urine, etc.), collection consumables may be sent to donors for a more accessible collection at home, being properly selected to guarantee the integrity of the samples until they are received by the SSPA Biobank. In the last case, instructions for handling samples and the informed consent form are also provided to donors, who are previously informed by the SSPA Biobank staff. The informed consent form for the donation of samples and associated data containing the information shown in Table 1 must be signed before each obtaining of samples. Samples will be processed, stored and/or distributed in a pseudonymized way by the SSPA Biobank according to the needs of the researchers after approval by the external scientific and ethics committees.

### 2.5. Database and Management System

The SSPA Biobank has implemented a global management system called nSIBAI, made up of different modules that allow the recording and monitoring of all the information associated with the biobank operations. This software has been co-developed by personnel from the Andalusian Public Health System and Biosoft Innovation S.L. The technology used for its development, the MongoDB database system, gives it great capacity and versatility to incorporate new functionalities and adapt the data model through a relatively simple configuration while maintaining the necessary security. In addition, it allows integrations with other software to facilitate access to additional information and speed up all the tasks performed. Currently, nSIBAI is working in more than 20 nodes of the SSPA Biobank, operating as a virtual biobank [8,9].

Within nSIBAI, a specific module has been developed for the Andalusian Registry of Donors for Biomedical Research (Figure 2). The data model has been designed in a standardized and normalized way in accordance with international initiatives on data harmonization [10,11]. Likewise, the recommendations of the ISBER (International Society for Biological and Environmental Repositories) on data collection and management have been taken into account [12]. This module allows you to record the following information:Personal, demographic and contact information completed in the application form. In order to minimize time consumption and possible errors, nSIBAI connects to the database of users of the Andalusian Public Health System through the national identification number or Andalusian medical record number of these donors.Structured and standardized information related to informed consent and its possible revocation, in addition to the form signed by the donor.Health data. The information provided by donors through the application form is validated and completed when necessary by accessing the medical records. To standardize this information, nSIBAI uses the International Classification of Diseases (ICD-10) as a reference. For healthy donors, a specific categorization has been designed in addition to that provided by the ICD-10.Epidemiological data such as consumption habits, activity data and other family or personal data of interest included in the application form. This information could be expanded in the future through new forms provided to donors.Information on relatives to record the existing relationship between donors incorporated into the Registry of Donors (mother–child, twins, etc.), especially relevant in some collections for the study of genetic pathologies.Information on whether donors belong to patient associations.Monitoring of the interaction of donors with the SSPA Biobank in relation to the communications and invitations received, inquiries made and donations to research projects.

The information is recorded by trained personnel of the SSPA Biobank who access nSIBAI with an exclusive and specific role, thus assuring high data security and confidentiality. Traceability of records and queries is an additional guarantee for donors’ privacy.

On the other hand, the Registry of Donors module that contains the donor database interacts with requests and projects, and nSIBAI sample and data management modules. Thanks to this innovative functionality, nSIBAI allows you to develop the actions shown in Figure 2:Search for registered donors from a specific research project included in nSIBAI, both through non-standard information provided by them and standardized IDC-10 information.Create and manage lists of donors potentially compatible with current research needs and their validation for the project.Associate information on collection, processing, storage and distribution of samples donated by donors incorporated into the Registry of Donors for specific research projects.Control panel of donors contacted to invite them to participate in the research project, and samples collected and distributed for each request.

## 3. Results 2015–2020

Since its creation in 2015, 1844 participants have been incorporated into the Andalusian Registry of Donors for Biomedical Research, increasing the number of donors annually until 2018 (Figure 3). It should be noted that the number of new donors is directly correlated with the promotion of the Registry, and these dissemination activities must be coordinated with other objectives of the SSPA Biobank and therefore may vary from year to year. In addition to the donors registered in the database, the SSPA Biobank has established framework agreements with patient associations regarding the promotion and dissemination of the Registry of Donors, collaboration in events and communication with the SSPA Biobank, having their members information available and the opportunity of participating in research projects in an abbreviated manner after the corresponding informed consent process. These donors are therefore registered at the time of donation, signing both informed consent forms, due to the fact the agreement does not imply direct access by the SSPA Biobank to the members’ data since contact is always established through the associations. However, the establishment of agreements with patient associations represents a very significant advance, in terms of the number of donors and, accordingly, the number of research projects attended. Currently, the SSPA Biobank has five effective national, regional or local agreements related to rare diseases, autoimmunity and pediatric or neurodegenerative pathologies, and it is planned to collaborate with twelve new patient associations in these pathologies and other conditions such as transplanted or disabled patients.

However, the Registry of Donors is not only for patients but also offers the opportunity for citizens (healthy, children, etc.) to get involved in the research process, both as donors and by promoting and providing added value. Thus, 60% are healthy donors when the distribution by disease of the registered participants is analyzed. On the other hand, the pathologies represented by the remaining 40% of donors are shown in Figure 4 and Figure 5, COVID-19 becoming the most representative pathology thanks to the specific campaign carried out throughout 2020, followed by neoplasms and allergies. The representation of rare diseases in the Register of Donors (9% donors, 57 different diseases) highlights the importance and opportunity of this initiative for research in this field. 

Furthermore, taking into account the total number of registered donors (1844), 68% were women compared to 32% men. The donor age distribution is shown in Figure 6, with 71% being between 20 and 50 years old.

Regarding the requests of investigators, the Registry of Donors has allowed 95 donors (5.1% of registered donors) to participate in 9 research projects (8 ended and 1 ongoing) since it became fully operational. A total of 220 primary samples of blood, urine, stool or saliva and associated data have been donated to these research projects related to autoimmune diseases, cancer, endocrine, nutritional and metabolic diseases and COVID-19, with the various objectives of identifying biomarkers for early diagnosis and response to treatment, validation of diagnostic tests and innovative procedures for analysis or quality control of samples. Despite the recent distribution of samples (83.6% since 2017), two articles are already available, although it will take more time to know the real impact of the Registry of Donors in terms of publications. Currently, six new projects pending sample distribution are being attended to.

## 4. Discussion

The Andalusian Registry of Donors for Biomedical Research was created in 2015 as a complementary and flexible tool to the classic SSPA Biobank operations to have a database available that contains the population willing to donate samples and associated information that meets specific preanalytical requirements for biomedical research. Before its constitution and operation, ethical–legal bases were carefully established and a specific informed consent form to incorporate into the Registry of Donors was elaborated, different and independent from the informed consent form to donate samples to the SSPA Biobank. Information included in both informed consent forms is shown in Table 1.

The informed consent form for the Registry of Donors sets the legal basis for data access from different sources (application form and medical records) and data processing that is needed for the operation of the Registry of Donors detailed in this manuscript and summarized in Figure 1. 

On the other hand, the informed consent form for the SSPA Biobank is mandatory before each obtaining of samples from donors included in the Registry. This consent allows donors to decide on the use of the donated samples specified in the form (one type or different types according to the needs of the projects) and the associated information, and to restrict the study areas based on non-welfare interests [13]. In addition, the SSPA Biobank is developing an interactive IT interface to give donors information about the use of their samples and whether these have been depleted and the research projects that have used the samples they donated, as well as to give and revoke consent to the use of their samples and associated data and to modify preferences over time, characteristics attributed to dynamic consent [14]. This interface offers donors control of how these samples are used and is taking into account the criticisms identified for dynamic consent [15].

The second important aspect addressed by the informed consent form for the SSPA Biobank is the possibility of contacting donors to collect additional data or samples or provide them with information relevant to their health (incidental findings). The guide published by BBMRI-NL [16] describes best practices for the detection, management and communication of incidental findings that have been taking into consideration as part of the procedure followed by the SSPA Biobank. Briefly, when incidental findings are reported by a researcher, the SSPA Biobank consults the informed consent form to find out if it is authorized to contact the donor and also the external ethics committee on the significance and consequences for the donor and their family members. If incidental findings can and should be communicated, the feedback is given by the treating physician. However, because of the advances in medical diagnostic technologies as well as the impacts and concerns associated with the return of incidental findings, more recent recommendations will be analyzed to establish a comprehensive framework [17].

Both informed consent forms were revised and approved by the external ethics committee of the SSPA Biobank. This committee was also consulted about the procedures of access to medical records necessary for the SSPA Biobank operation in order to establish best practices and ethical guidelines as it has been suggested in [18]. The informed consent forms are also valid for minors, whose legal guardian has to sign and inform them about their incorporation into the Registry of Donors and/or donation of samples and their rights. As supporting material for minors during the informed consent process, a comic with the description of the Registry of Donors is used. When minors reach the majority age, the SSPA Biobank contacts them in case of sample collection to confirm their participation in the Registry of Donors.

The management of donors’ dispositions related to consent are facilitated by the software nSIBAI, made up of different modules including the specific one for the Registry of Donors, and that all together allow the recording and monitoring of all the information associated with the biobank operations. Thanks to this software, the SSPA Biobank will face challenges and problems existing for current data sharing and management systems [19]. Of importance, the Registry of Donors can currently incorporate donors from other regions different from Andalusia through a comprehensive health questionnaire to be recorded in nSIBAI. Additionally, more sophisticated interactions with medical records or connections to other donor registries can be established through new collaborations to increase the reach on this tool.

The Andalusian Registry of Donors for Biomedical Research allows societal actors to be involved in the entire research process, establishing alliances with patient associations and groups to develop joint actions and promote biomedical research. The citizen demand for active participation in research was precisely the starting point for the creation of the Registry of Donors, which has designed different strategies to avoid bias in recruitment pointed out by some authors [20]. On the other hand, protecting the rights of citizens who donate their samples for research was the motivation for the creation of the SSPA Biobank. Thus, donors are a fundamental stakeholder of the SSPA Biobank, which continuously works to generate transparency and trust. Promotion and dissemination activities on the handling of biological samples for research and the operation of the SSPA Biobank, as well as the Registry of Donors, the interactive IT interface described in this manuscript and the incorporation of principles for stakeholder engagement [21] as part of SSPA Biobank processes will contribute to this objective.

## 5. Conclusions

The Andalusian Registry of Donors for Biomedical Research is an innovative and powerful initiative for the recruitment of donors and the prospective collection of samples for research that engage patients and citizens interested in donating samples for research. During its first five years of life, ethical–legal bases were consolidated, operation and management were established and processes were certified by the ISO 9001 standard. Next, the SSPA Biobank plans to expand its scope and collaborations through national and international projects in order to raise the impact of the Registry of Donors on research. In this sense, dissemination activities are essential to increase the participation of donors, but also the demand from researchers; the SSPA Biobank will work to reach this objective. 

## Figures and Tables

**Figure 1 biotech-10-00006-f001:**
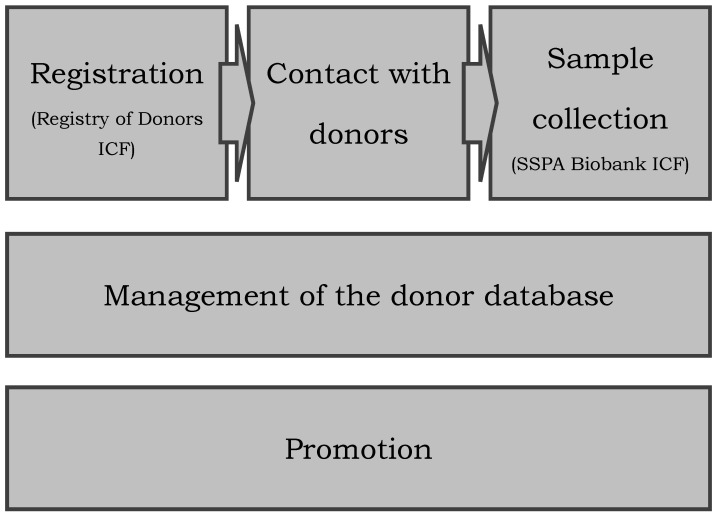
Main processes of the operation of the Andalusian Registry of Donors for Biomedical Research. ICF, informed consent form.

**Figure 2 biotech-10-00006-f002:**
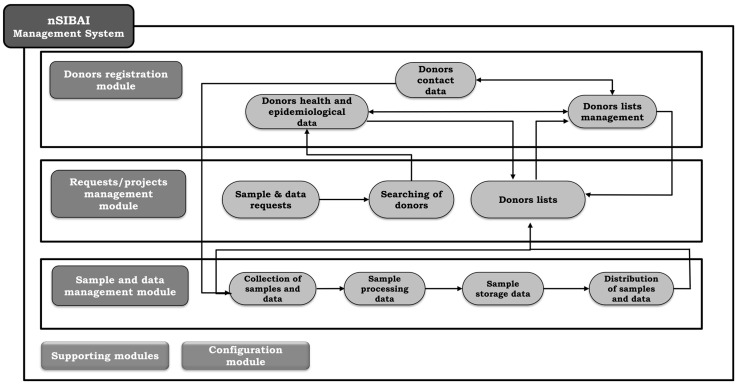
Interaction of the nSIBAI modules including the specific module for the Andalusian Registry of Donors for Biomedical Research.

**Figure 3 biotech-10-00006-f003:**
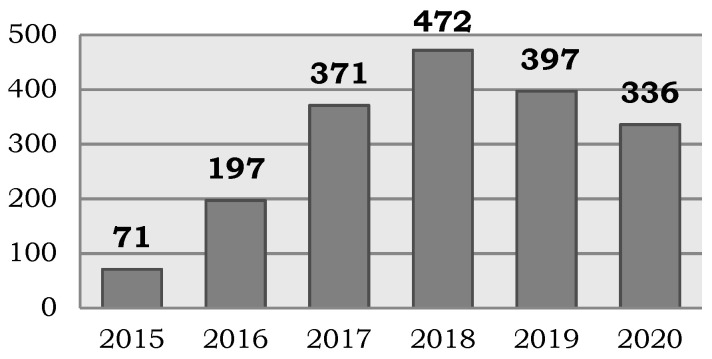
Number of donors annually incorporated into the Andalusian Registry of Donors for Biomedical Research since its creation in 2015.

**Figure 4 biotech-10-00006-f004:**
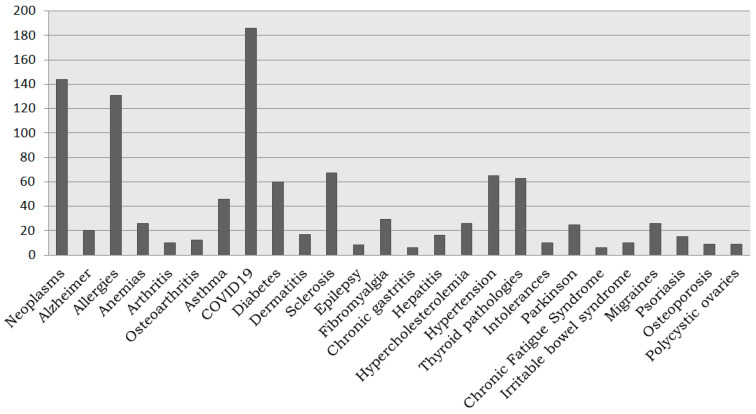
Most representative pathologies in the Andalusian Registry of Donors for Biomedical Research.

**Figure 5 biotech-10-00006-f005:**
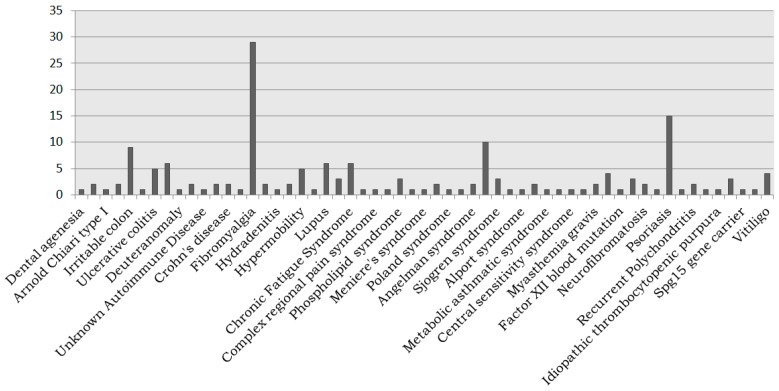
Presence of rare diseases in the Andalusian Registry of Donors for Biomedical Research.

**Figure 6 biotech-10-00006-f006:**
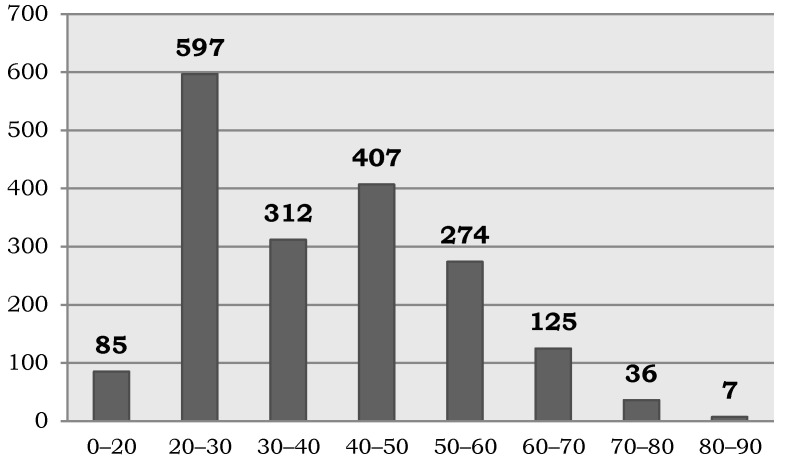
Distribution by age of the donors incorporated into the Andalusian Registry of Donors for Biomedical Research since its creation in 2015.

**Table 1 biotech-10-00006-t001:** Information included in the application form (A) and informed consent form of the Andalusian Registry of Donors for Biomedical Research (B), as well as in the informed consent form of the SSPA Biobank (C).

(A) Application Form for the Registry of Donors	(B) Informed Consent Form for the Registry of Donors	(C) Informed Consent Form for the SSPA Biobank
Name	Description of the Registry	Description of the SSPA Biobank
Gender	Description of the SSPA Biobank	Regulations regarding donation for research
National identification number	Data protection guarantees	Obtaining and use of samples
Communications address	Possibility of withdrawing the consent	Information available about the use of samples and results of research
Phone number	**Consent to:**	Possibility of being contacted by the SSPA Biobank
Email address	Be part of the Registry	Data protection guarantees
Date of birth	Data processing	Possibility of withdrawing the consent
Clinical condition/disease	Be contacted by the SSPA Biobank	Genetic analysis
Date of diagnosis	Access to medical record	**Consent to:**
Andalusian medical record number		Donate samples and data
Epidemiological information		Use of samples and data for selected research areas and restrictions
		Be contacted by the SSPA Biobank
		Receive information about incidental findings and genetic analysis

## Data Availability

The data are not publicly available due to privacy and ethical restrictions.

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
