# Peer review of "The Andalusian Registry of Donors for Biomedical Research: Five Years of History"

_biotech, 2021, doi:10.3390/biotech10010006_

Round 1
Reviewer 1 Report
Generally speaking, this article does present to readers a new and innovative model of biobanking, which can be taken as a good contribution, both in terms of academic and practical senses, to the future biomedical research in the Big Data Era. However, trying to make an innovation be sustainable, a more comprehensive coverage of basic ingredients of a successful biobank become necessary, too. Listed below are some of the questions raised for the authors' consideration. 1. On line 75, "donor medical records such the bone marrow registry were used as a reference for the," would that be better to replace "such" with "such as"? 2. For the biobank at stake is an European one, it might be necessary to also include some GDPR compliance descriptions; i.e., the new rights granted to participants; such as rights to access, rights to restrict processing, rights to be forgotten, rights to erasure, rights to withdrawal & etc. The authors might want to add on these somewhere between lines 76-79. 3. In regard to the retroactive collection by utilizing donors' information described on lines 154-155, this tends to be a practice possibly in conflict with the general privacy protection requirement to avoid "re-identification, the authors might want to supplement on this and explain the management setting and/or legal basis to justify the practice. 4. On line 161, it seems that another appointment will be set for a new participation, would there be a new consent required? Or, it's been included in the original broad consent? 5. in line 174, the text refers only to "non-intrusive" part. How about the rest? 6. The texts between 253-256 did present to us a very effective and efficient design. On the other hand, it becomes necessary for the authors to also clarify the shadow of privacy infringement might be casted on those participants of sensitive and/or rare disease groups.Author Response
Please see the attachment.

Reviewer 2 Report
This is an interesting though largely descriptive piece about a specific initiative. A number of minor changes would be helpful before publication.
The main one is that it would be desirable at least to address some possible ethical concerns about privacy. The section on page 6 between lines 221-5 is very unclear about the privacy implications, and more could usefully be said about this.
This also applies to te point about 'information on relatives' mentioned on page 6 line 214.
On page 5 line 164-5 it is stated that if people are not willing it is recorded. Is the reason recorded?
There are several places where there are typographical errors or where the choice of words is somewhat unusual or strange:
Page 2 line 75 'as' missing after 'such'
line 79. 'withdrawal' would be a more normal term than 'retirement'. I note that at line 206 the word ' revocation' is used, which is also fine.
Page 3 line 95. 'attractive' What does this mean? Attractive to whom?
Page 3 line 111. 'This was the case of' It would be more usual to say ' This was the case with...'
Page 4. line 137. 'have' should be 'has'
Page 5. line 160 'associated data with donating' This is a bit unclear. Better 'data that will be associated with donating'
Page 5 line 175 'obtaining consumables' - what are these?
Page 7. line 249 'abbreviate' sounds odd. Better: ' abbreviated'
Reviewer 3 Report
The paper describes a study concerning different aspects of donor Biobank in Andalusia -Spain.
The study is well conducted and well described, however it needs deeper analysis of some aspects like information to the donor and consent, and the aspect of the relationship with the citizens.
The results paragraph needs an implementation with a brief description of the research projects that have been conducted or that are now in progress, to demonstrate the usefulness of this Biobank: maybe some research projects have already produced some results that have been published? It would be nice to add some indication of papers that present research on samples obtained from Registry of donors, if available.
Furthermore I think that an implementation of the references is needed.
I suggest the following revisions:
- In general, the paper need an implementation of the discussion and the references (due to the huge amount of literature that has been published on the topic) regarding how the informed consent of the donor to a research biobank is organized and managed. In the case described in this paper does the consent regard only one type of samples or also research that involve different samples? It is a model of dynamic consent ora broad consent? How is the information to the donor structured? If the research will be expanded would the biobank re-contact the donors? Have you consider a possible role of the ethics committee?
I also have seen that in the donors included there is the category 0-20, this means that you have a number of minors. How you managed the informed consent of minors? You recontact them after the majority age to confirm the donation?
I suggest to implement with references, for example, as follows, considering also to add other that you will find in the flourishing literature on biobanks.
-Gefenas E, Dranseika V, Serepkaite J, Cekanauskaite A, et al. Turning residual human biological materials into research collections: playing with consent. J Med Ethics. 2012 Jun;38(6):351-5.
- Caenazzo L, Tozzo P, Borovecki A. Ethical governance in biobanks linked to electronic health records. Eur Rev Med Pharmacol Sci. 2015 Nov;19(21):4182-6.
- Budin-Ljøsne I, Teare HJ, Kaye J, Beck S, et. al. Dynamic Consent: a potential solution to some of the challenges of modern biomedical research. BMC Med Ethics. 2017 Jan 25;18(1):4.
- Langhof H, Schwietering J, Strech D. Practice evaluation of biobank ethics and governance: current needs and future perspectives. J Med Genet. 2019 Mar;56(3):176-185.
- Pacyna JE, McCormick JB, Olson JE, et al. Assessing the stability of biobank donor preferences regarding sample use: evidence supporting the value of dynamic consent. Eur J Hum Genet. 2020 Sep;28(9):1168-1177.
-Prictor M, Lewis MA, Newson AJ, Haas M, et al.. Dynamic Consent: An Evaluation and Reporting Framework. J Empir Res Hum Res Ethics. 2020 Jul;15(3):175-186.
2) The discussion paragraph regarding the promotion of the registry of donors in the society, could be implemented with a general discussion on the relationship between citizen and biobanks, regarding, for example, trust, engagement, participation, and the efforts that the biobank put on these issues. See for example:
- Labarga A, Beloqui I, Martin AG. Information Management. Methods Mol Biol. 2017;1590:29-39.
- Prictor M, Teare HJA, Kaye J. Equitable Participation in Biobanks: The Risks and Benefits of a "Dynamic Consent" Approach. Front Public Health. 2018 Sep 5;6:253.
-Tozzo P, Caenazzo L. The Skeleton in the Closet: Faults and Strengths of Public Versus Private Genetic Biobanks. Biomolecules. 2020 Sep 3;10(9):1273. doi: 10.3390/biom10091273. PMID: 32899386; PMCID: PMC7564942.
- Schmanski A, Roberts E, Coors M, et al. Research participant understanding and engagement in an institutional, self-consent biobank model. J Genet Couns. 2021 Feb;30(1):257-267.
-Mezinska S, Kaleja J, Mileiko I. Becoming and being a biobank donor: The role of relationships and ethics. PLoS One. 2020 Nov 23;15(11):e0242828.
- Eklund N, Andrianarisoa NH, van Enckevort E, Anton G, et al. Extending the Minimum Information About BIobank Data Sharing Terminology to Describe Samples, Sample Donors, and Events. Biopreserv Biobank. 2020 Jun;18(3):155-164.
3) The results paragraph needs an implementation with a brief description of the research projects that have been conducted or that are now in progress, to demonstrate the usefulness of this Biobank.
4) Is there any procedure to manage incidental findings and the return of actionable results to the donors? If yes, it would be good to describe it. This is an aspect that sometimes is not usually considered, but I think that it would improve your manuscript.
- De Clercq E, Kaye J, Wolf SM, Koenig BA, Elger BS. Returning Results in Biobank Research: Global Trends and Solutions. Genet Test Mol Biomarkers. 2017 Mar;21(3):128-131.
- Lin JC, Hsiao WW, Fan CT. Managing "incidental findings" in biobank research: Recommendations of the Taiwan biobank. Comput Struct Biotechnol J. 2019 Aug 2;17:1135-1142.
-Thorogood A, Dalpé G, Knoppers BM. Return of individual genomic research results: are laws and policies keeping step? Eur J Hum Genet. 2019 Apr;27(4):535-546.
5) Reference n.3 is not complete, please specify all the Authors (avoid et. al)
6) Change in the title the number 5 with “five” in word.
